# Health-Related Quality of Life in Stroke Survivors in Relation to the Type of Inpatient Rehabilitation in Serbia: A Prospective Cohort Study

**DOI:** 10.3390/medicina56120666

**Published:** 2020-11-30

**Authors:** Natasa K. Rancic, Milan N. Mandic, Biljana N. Kocic, Dejan R. Veljkovic, Ilija D. Kocic, Suzana A. Otasevic

**Affiliations:** 1Faculty of Medicine Nis, University of Nis, 18000 Nis, Serbia; biljaizzz@yahoo.com (B.N.K.); ilija.k@hotmail.com (I.D.K.); otasevicsuzana@gmail.com (S.A.O.); 2Institute for Public Health Nis, 18000 Nis, Serbia; 3Clinic for Physical Medicine and Rehabilitation, Clinical Center Nis, 18000 Nis, Serbia; milanmandic69@gmail.com; 4Ministry of Internal Affairs, Gendarmerie Detachment in Kraljevo, 36000 Kraljevo, Serbia; drdejanveljkovic@gmail.com

**Keywords:** stroke, inpatient rehabilitation, additional rehabilitation, health-related quality of life

## Abstract

*Background and objectives*: Health-related quality of life after stroke is an important public health issue. The objective of the study was to investigate the relationship between the perceived health-related quality of life in stroke survivors in relation to the type of inpatient rehabilitation. *Materials and Methods*: Using a random selection method out of a total of 688 patients, every fourth survivor who had a stroke in the period from 1 January 2017 to 31 December 2019 was selected from the admission protocol of the Clinic for Rehabilitation and Physical Medicine of the Clinical Centre Niš, Serbia. A total of 160 first-ever stroke survivors were included (80 underwent additional inpatient rehabilitation and 80 underwent only inpatient rehabilitation in a tertiary health institution) in a twelve-month prospective study. The EuroQuol-5 Dimension (EQ5D) questionnaire and Stroke Impact Scale were used for the assessment. Multivariate linear regression analysis was done. *Results*: Multivariate linear regression analysis showed that additional inpatient rehabilitation from six up to eight weeks after discharge was significantly associated with better self-reported health condition by 3.9 times (from 1.9 to 8.2), significantly decreased the ranks of EQ5D by 1.78 times (from 1.02 to 3.11), and showed a higher health-related quality of life. We determined a significant increase of strength, emotions, mobility, and participation role in survivors who underwent additional inpatient rehabilitation compared with those who did not. *Conclusions*: There was a significant difference in health-related quality of life perceived by stroke survivors who underwent additional hospital rehabilitation in relation to those who underwent only inpatient rehabilitation.

## 1. Introduction 

Stroke is the second most common cause of premature death and the leading cause of significant adult disability [1,2]. More than 67.5 million people are currently living who have experienced an ischemic stroke and about 61% of them are under the age of 70 [3]. Women represent over a half (51%) of all survivors living with the consequences of stroke [4,5].

The definition of the health-related quality of life (HRQOL) refers to the aspects of the quality of life affected by a disease, or the impact of the health condition or health care intervention on the individuals’ subjective experience in functional, cognitive, social, and psychological processes [6]. 

HRQOL measurements are potentially more relevant to patients than measures of impairment or disability and are an important index of outcome after stroke that can facilitate a broader description of the disease and outcomes [7]. Impaired and decreased HRQOL is the most important consequence of stroke for stroke survivors [8]. 

More efficient treatment of stroke in acute phase and significant reductions in hypertension, tobacco smoking, increased use of anticoagulants for atrial fibrillation [9], as well as the fact that the majority of strokes is not fatal [2,3] have led to an increase in the number of stroke-disabled survivors [4,5]. A decrease in stroke mortality by 70% in the United States of America (USA) is recognized as one of the 10 great public health achievements of the 20th century [9]. 

It has been estimated that of those individuals who survive a stroke, only 65% may be functionally independent one year following the stroke event [10]. Both physical and mental disabilities are common after stroke [11]. It is assessed that only 10% of survivors make a full recovery, about 25% of all survivors recover with minor impairments, and about 40% of all survivors continue to live with moderate disabilities, while 15% to 30% live with severe disabilities and are fully dependent. About 25% of survivors need additional treatment and rehabilitation in specialized institutions [12,13]. 

The population of stroke survivors is the largest population of patients in inpatient rehabilitation (IR) facilities [14]. 

In Serbia, stroke survivors with a functional deficit after acute treatment of stroke in intensive care unit (ICU) at the Clinic of Neurology are offered 14 to 30 days of inpatient rehabilitation in stationary health institutions. Rehabilitation begins in the ICU when the patient’s state is stable and after discharge from ICU. The majority of survivors continue rehabilitation at the stationary health institutions where they get additional IR. An inpatient interdisciplinary stroke team decides about the mode of rehabilitation.

The individual factors that significantly influenced stroke-specific HRQOL were the social participation and functional status [15]. Gurcay et al. also found that both functional status and age had a powerful influence on the HRQOL in stroke survivors. Improving functional disability, particularly s associated with higher HRQOL [16].

It was previously considered that positive effects of rehabilitation treatment can be achieved only during the first three to six months after the stroke [17,18,19]. If cognitive functions are maintained after the stroke, continuous rehabilitation can improve independence, recovery of affected functions, and the quality of life even several years after the stroke [20,21,22,23,24,25]. 

It is not yet clear whether additional rehabilitation in the early period of recovery could increase HRQOL and how long the achieved positive effects would last [8,13,14]. 

The main objective of the study was to investigate the relationship between the perceived health-related quality of life in stroke survivors in relation to the type of inpatient rehabilitation. 

## 2. Material and Methods

### 2.1. Study Design

A total of 2199 patients were admitted to the Clinic for Rehabilitation and Physical Medicine of the Clinical Center Nis, Southeast Serbia, for inpatient rehabilitation, from 1 January 2012 up to 31 December 2019.

The participants for the study were selected by using a random sampling method out of a total of 688 patients where every fourth survivor who had a stroke in the period from 1 January 2017 to 31 December 2019 was selected from the admission protocol of the Clinic for Rehabilitation and Physical Medicine, of the Clinical Center Nis, Southeast Serbia. The number of participants thus selected was 172.

We conducted a prospective cohort study which involved 172 stroke survivors who had first-ever stroke in the period from 1 January 2017 to 31 December 2019, and who fulfilled the inclusion criteria.

The inclusion criteria were first-ever stroke, age from 30 to 79 years, survivors from the territory of Southeast Serbia, possible communication with participants, and the written consent of all participants to participate in the study.

The exclusion criteria were previously experienced stroke, insufficient cooperation of patients, psycho-organic syndrome, aphasia, transient ischemic attack (TIA), new stroke within 90 days of the first stroke, complications after stroke, and death of the participant.

Out of the total number of 172 stroke survivors, only 160 completed the study. Dropping out was caused by the following: 1 survivor declined to participate, 7 had recurrent stroke, and 4 died. 

Out of 160 stroke survivors, 80 underwent additional inpatient rehabilitation (IR). First, they had IR for a period of 30 days at a tertiary health institution, Clinic for Physical Medicine and Rehabilitation of the Clinical Centre Nis, in Southeast Serbia. Additional IR was administered in an early sub-acute phase (from 6 up to 8 weeks after discharge from the Clinic for Physical Medicine and Rehabilitation). The survivors received additional IR in a state-owned tertiary health institution, Institute for Prevention, as well as treatment and rehabilitation of cardiovascular and rheumatoid patients "Radon" in a spa near the city of Nis. It included physiotherapy procedures, aerobic exercises, and walking under medical supervision.

Eighty survivors underwent only IR for 30 days at the Clinic for Physical Medicine and Rehabilitation of the Clinical Centre Nis, Southeast Serbia. 

All survivors were transferred to the Clinic for Physical Medicine and Rehabilitation on the same day they were discharged from the intensive care unit (ICU) at the Neurology Clinic. They received specialized interventions, kinesitherapy, electrotherapy, and occupational therapy. Stroke was diagnosed by a neurologist when the patient was hospitalized at the ICU at the Neurology Clinic Nis. Computer Tomography (CT) and Magnet Radio Imaging (MRI) were used in diagnostic, too.

Values of the HRQOL 4 weeks/one month after discharge from the IR were considered baseline values, and values at six and twelve months after discharge were considered control measuring. 

Basic demographic data and stroke-related characteristics (age, gender, marital status, income, type of stroke, stroke localization) were obtained from medical records. We also assessed the HRQOL six and twelve months after the first assessments using standardized generic and specific self-administered questionnaires. Trained researchers and physicians distributed questionnaires to the survivors in their homes after discharge from IR.

All participants in the study were informed in detail about the objectives of the study and they all signed the consent to participate in the research. 

The approval for the study was obtained from the Ethics Committee of the Clinical Center Nis (number of Decision: No. 2280/12, 1 February 2012).

### 2.2. Questionnaires 

We assessed HRQOL by commonly used standardized questionnaires, Serbian version of generic EuroQuol 5 Dimension (EQ5D) and specific Stroke Impact Scale (SIS).

### 2.3. Euro Qol 5 Dimensions (EQ5D)

The EuroQol-5 Dimension is a standardized measure of health status developed by the Euro QoL Group in order to provide a simple, generic measure of health for clinical and economic appraisal. EQ-5D is designed for self-completion by respondents and is ideally suited for use in postal surveys, in clinics, and in face-to-face interviews. The EQ-5D descriptive system comprises the following 5 dimensions: mobility, self-care, usual activities, pain/discomfort, and anxiety/depression. Each dimension has 3 levels: no problems, some problems, severe problems. The respondent is asked to indicate his/her health state by ticking (or placing a cross) the box against the most appropriate statement for each of the 5 dimensions. This decision results in a 1-digit number expressing the level selected for that dimension. The digits for 5 dimensions can be combined in a 5-digit number describing the respondent’s health state. It should be noted that the numerals 1–3 have no arithmetic properties and should not be used as a cardinal score [26]. 

The EQ Visual Analogue Scale (EQVAS) records the respondent’s self-rated health on a vertical, visual analogue scale where the endpoints are labeled ‘best imaginable health state’ and ‘worst imaginable health state’. This information can be used as a quantitative measure of health outcome as judged by individual respondents [26]. 

### 2.4. The Stroke Impact Scale (SIS)

The Stroke Impact Scale (SIS) is a stroke-specific and self-reported questionnaire which was developed by Duncan et al. [27] and it consists of 59 items measuring eight domains (strength, hand function, activities of daily living/instrumental activities of daily living, mobility, communication, emotion, memory and thinking, and participation role). Each domain of SIS has a range of 0–100 and higher scores indicated better HRQOL.

### 2.5. The Statistical Analysis

All the calculations were done by the SPSS software package version 20.0 (IBM Corp., Armonk, NY, USA) and S-PLAS program version 2000 (TIBCO Software Inc., Palo Alto, CA, USA) Student’s *t*-test was used to compare numerical differences of normal distribution and the Mann–Whitney U test was used to compare two values when the dependent variable is either ordinal or continuous, but not normally distributed, and the Chi square test was used, too. 

Multivariate stepwise logistic regression analysis (Enter method) was used. A *p*-value below 0.05 (*p* < 0.05) was considered statistically significant. 

## 3. Results

A total of 160 stroke survivors (80 underwent additional IR and 80 underwent only IR) completed the study. The average age of the survivors with additional IR was 60.94 ± 8.74, and of those with only IR, it was 62.30 ± 9.43 on average. Survivors in both groups were predominantly female (more than 60%), married (more than 80%), and with below average income (more than 85%) (Table 1). 

No significant difference was found between the survivors according to the age, gender, marital status, and income. 

There was no significant difference in the type of stroke distribution and localization between survivors with different type of rehabilitation (*p* > 0.05) (Table 1).

The basic descriptive characteristics of the survivors and stroke-related characteristics are shown in Table 1. 

There were considerably more stroke survivors who had significantly higher BMI (*p* = 0.001), higher blood pressure (*p* = 0.05), and more prevalent diabetes mellitus (*p* = 0.04) in the group who underwent only IR compared with those who underwent additional IR (Table 1).

There was a similar distribution of other conditions (risk factors and chronic diseases) in survivors and the difference was not significant (*p* > 0.05).

After discharge from the IR, the average values of EQVAS were the lowest in both groups of survivors. Six and twelve months after discharge, the average values of EQVAS in relation to the baseline values significantly increased both in males (*p* < 0.001) and in females (*p* < 0.001).

After six and twelve months, the average values of EQVAS were significantly higher in males than in females (Table 2).

Six (*p* < 0.05) and twelve months (*p* < 0.001) after discharge from the IR, the average values of EQVAS were significantly higher in survivors who underwent additional IR compared with those who underwent only IR (Table 3). 

A significant decrease of EQ5D ranks in both groups of survivors was observed in the first six months after discharge from the IR. During the period of six to twelve months, a decrease of ranks was determined only in survivors who underwent additional IR (Table 3).

Average values of EQ5D domains in the period from six to twelve months are shown in Table 4.

According to data presented in Table 4, all EQ5D domains improved in both groups of survivors and the determined difference was not significant. Only the domain Mobility significantly improved in those survivors who underwent additional IR.

The average scores of the SIS domains baseline, six and twelve months after discharge from the Clinic for Physical Medicine and Rehabilitation in survivors who underwent only IR rehabilitation vs survivors who underwent additional IR, are shown in Table 5.

Baseline average scores in both groups of survivors for the strength, mobility, ADL, hand function, and participation role domains were the most decreased. The highest average scores in this period were found for communication, memory, and emotion domains.

Six months after discharge from IR, the domain scores were under 50 in both groups of survivors for mobility, ADL, hand function, and participation role.

Six and twelve months after discharge, we determined a significant increase of mobility; after 6 months (*p* = 0.018) and after 12 months (*p* = 0.048) in participation role; after 6 months (*p* = 0.005) and after 12 months (*p* = 0.008) in survivors with additional IR compared with those who underwent only IR (Table 5).

Twelve months after discharge, we determined a significant increase of the strength (*p* = 0.046) and emotions (*p* = 0.027) domains in survivors with additional IR compared with those who underwent only IR (Table 6).

Effects of IR and additional IR on the average values of EQVAS and the ranks of EQ5D are presented in Table 6.

Additional IR significantly increases the average values of EQVAS in survivors in the period from 6 to 12 after discharge by 3.9 times (from 1.9 to 8.2).

Additional IR significantly decreases the ranks of EQ5D in the period from 6 to 12 after discharge by 1.78 times (from 1.02 to 3.11) and it increases the HRQOL.

## 4. Discussion

Despite the fact that nine out of ten strokes could be prevented, stroke is still the second most common cause of significant disability of the adult population worldwide [1] and it is one of the biggest public health issues today. Stroke affects all physical and psychosocial domains of quality of life of survivors [28]. Rehabilitation procedures applied early after stroke in health institutions under medical supervision can significantly improve the functional outcomes and health-related quality of life for stroke survivors.

In this twelve-month prospective study, we compared the differences of HRQOL in stroke survivors who underwent additional inpatient rehabilitation at a spa under medical supervision after inpatient rehabilitation in a rehabilitation facility with those who underwent only inpatient rehabilitation in a rehabilitation facility. The lowest HRQOL in both groups of stroke survivors was one month after discharge from the inpatient rehabilitation in a rehabilitation facility. The first measurement showed the lowest perception of health condition and the highest ranks of EQ5D, which indicates the lowest HRQOL. The most decreased SIS domains were strength, mobility, hand function, ADL, and participation role.

Our findings showed a significant decrease of EQ5D ranks in both groups of survivors in the first six months after discharge; and in the period from six to twelve months, a decrease of EQ5D ranks and increase of HRQOL was determined only in survivors who underwent additional IR.

The results of multivariate logistic regression analysis showed a significant relationship between the type of rehabilitation and better estimation of health condition as well as higher HRQOL. The survivors who underwent additional IR conducted at a spa had 3.9 times better perception of their health condition and higher HRQOL. Women and older survivors had lowed HRQOL than men and younger survivors, but the difference is not statistically significant.

The results of our study are in agreement with the results of other studies There are similar results in the literature [7,18,19,22,23,29].

Six months after discharge from IR, we found that survivors who underwent additional IR had significantly higher mobility and participation role domains in the SIS questionnaire compared with those who had only IR. Twelve months after discharge from IR, the strength and emotion SIS domains were significantly higher. Other domains increased but there were no significant differences between the groups of survivors.

According to the data from one study that assessed 63 stroke survivors during inpatient rehabilitation, one month after the stroke and again at home six months after the stroke, it was found that functional independence and HRQOL improved over time, but this improvement was strongly correlated with self-care and self-efficacy [28].

A study of HRQOL in 99 stroke survivors by EQ5D questionnaires six weeks, three months, and twelve months after the stroke onset showed that after twelve months of follow-up, there were significantly higher values of mobility, self-care, anxiety, and bodily pain compared to six weeks and three months after the stroke onset [20].

HRQOL was assessed using the Stroke Impact Scale (SIS), among two hundred twenty-nine participants three to nine months post stroke. Poorer HRQOL in the physical domain was associated with age, nonwhite race, more comorbidities, and reduced upper-extremity function [21].

Mandic et al. evaluated the functional outcome and HRQOL in 136 stroke survivors who underwent inpatient rehabilitation one, three, and six months. Six months after discharge from IR, all domains of the SF-36 questionnaire improved, except bodily pain [7].

Chen et al. analyzed nine studies of HRQOL in stroke survivors in a meta-analysis and they determined that the highest recovery was in the first month after stroke and no significant improvement was noted later [22].

An investigation of functional and motor recovery of upper limbs in stroke survivors showed that the higher functional improvement of mild motor dysfunctions was achieved in the first six weeks after the stroke [17].

Antic assessed HRQOL in stroke survivors and found that average scores for physical, social, and mental domains were almost identical. In the 6–12 months period, there was a slight improvement of physical domain compared with one month before stroke. There was no improvement in social functioning, mental health, vitality, and decease of bodily pain, compared with one month after the stroke onset [23].

Suinkeler evaluated HRQOL by the SF-36 questionnaire three, six, and twelve months after stroke and determined that twelve months after stroke, 66% of participants assessed their HRQOL as worse than before stroke. Physical and social functions significantly decreased and mental function significantly increased in the 6- to 12-month period [29].

Stavem assessed the HRQOL during six months after the stroke onset and there was no significant improvement in all HRQOL dimensions despite the applied additional rehabilitation. Only physical and mental components improved [30].

Anderson et al. assessed the HRQOL three and twelve months after the stroke. They found lower HRQOL in survivors with moderate disabilities after one and three months, especially in domains of physical function, physical role, emotional role, social function, and general health [31].

An observational cohort study of 1195 patients which was carried out from 17 February 2015 to 27 January 2017 showed that patients with ischemic stroke reported symptoms in multiple domains and physical function, satisfaction with social roles, and executive function were most affected [32].

## 5. Conclusions

There was a significant difference in the health-related quality of life perceived by stroke survivors who underwent additional hospital rehabilitation in relation to those who underwent only inpatient rehabilitation. Four domains of the specific SIS questionnaire significantly increased and EQ5D ranks significantly decreased after twelve months of follow-up only in survivors who underwent additional inpatient rehabilitation. Further studies about additional inpatient rehabilitation are needed.

## Figures and Tables

**Table 1 medicina-56-00666-t001:** Descriptive and stroke-related characteristics of the survivors in examined groups.

Characteristics	Survivors		Test and *p*-Value
	With Additional IR (*n* = 80)	Without Additional IR (*n* = 80)	
Average age	60.94 ± 8.74	62.30 ± 9.43	ns
Sex—female	48 (60%)	51 (63.8%)	ns
**Income status**			
Below average	71 (85.5 %)	70 (87.5%)	0.145
Average	9 (14.5%)	10 (15.0%)	0.419
Above average	0 (0.0%)	0 (0.0%)	0.0
**Marital status**			
Married	68 (85.0%)	70 (87.5%)	ns
Single	12 (15.0%)	10 (12.5%)	ns
**Stroke type**			
Ischemic	61 (76.3%)	62 (77.5%)	0.126
Hemorrhagic	19 (23.7%)	18 (22.5%)	
**Localization of stroke**			
Right hemisphere	43 (53.5%)	45 (56.2%)	0.750
Left hemisphere	17 (21.6%)	19 (23.8%)	0.068
Both hemisphere	4 (5.0%)	4 (5.0%)	0.0
Other localizations	16 (20.0%)	12 (15.0%)	0.224
**Risk factors**			
Body Mass Index	25.37 ± 2.33	27.09 ± 3.63	§ *t* = 3.30, *p* = 0.001
High blood pressure	44 (55.0%)	56 (70.0%)	* χ^2^ = 3.82, *p* = 0.05
Diabetes mellitus	16 (20.0%)	24 (30.0%)	* χ^2^ = 4.08, *p* = 0.04

IR—Inpatient Rehabilitation; § Student’s *t*-test; * Chi squared test, ns—non significant.

**Table 2 medicina-56-00666-t002:** Average values of the EQ Visual Analogue Scale (EQVAS) by sex, time, and type of inpatient rehabilitation.

Survivors	Sex	One Month after Discharge (Baseline Value)	Six Months after Discharge	Twelve Months after Discharge	Comparison by Time
With additional IR (*N* = 80)	Male	48.61 ± 11.01	68.98 ± 9.68	76.76 ± 11.62	A ^‡^, B ^‡^, C ^‡^
	Female	45.77 ± 11.29	68.08 ± 9.81	75.38 ± 11.04	A ^‡^, B ^‡^, C ^‡^
	Comparison	ns	ns	ns	
Without additional IR (*N* = 80)	Male	42.75 ± 11.10	51.47 ± 6.19	63.14 ± 11.57	A ^‡^, B ^‡^, C ^‡^
	Female	45.31 ± 11.19	53.79 ± 10.15	60.00 ± 9.64	A ^†^, B ^‡^, C ^‡^
	Comparison	ns	ns	ns	

A, one month after vs six months after discharge; B, one month after vs twelve months after discharge; C, six months after discharge vs twelve months after discharge. ^†^
*p* < 0.01; ^‡^
*p* < 0.001.

**Table 3 medicina-56-00666-t003:** The average ranks of the EuroQuol-5 Dimension (EQ5D) by type of inpatient rehabilitation, sex, and time of assessment.

Survivors	Sex	One Month after Discharge (Baseline Value)	Six Months after Discharge	Twelve Months after Discharge	Comparison by Time
With additional IR (*N* = 80)	Male	2.24 ± 0.24	1.43 ± 0.27	1.38 ± 0.28	A ^‡^, B ^‡^, C ^‡^
	Female	2.07 ± 0.28	1.45 ± 0.23	1.38 ± 0.29	A ^‡^, B ^‡^, C ^‡^
	Comparison	ns	ns	ns	
Without additional IR (*N* = 80)	Male	2.15 ± 0.31	1.59 ± 0.22	1.47 ± 0.25	A ^‡^, B ^‡^, C ^‡^
	Female	2.12 ± 0.30	1.67 ± 0.25	1.45 ± 0.23	A ^†^, B ^‡^, C ^‡^
	Comparison	ns	ns	ns	

A, one month after discharge vs six months after discharge; B, one month after discharge vs twelve months after discharge; C, six months after discharge vs twelve months after discharge. ^†^
*p* < 0.01; ^‡^
*p* < 0.001.

**Table 4 medicina-56-00666-t004:** Average values of EQ5D domains from six to twelve months.

Six Months after Discharge	Mobility	Self-Care	Usual Activities	Pain/Discomfort	Anxiety/Depression
Additional IR	1.28 ± 0.54	2.07 ± 0.67	2.12 ± 0.65	2.02 ± 0.71	2.06 ± 0.70
Only IR	1.73 ± 0.68	2.04 ± 0.70	2.04 ± 0.74	2.03 ± 0.73	2.09 ± 073
	*p* < 0.001	ns	ns	ns	ns
**12 Months after Discharge**	**Mobility**	**Self-Care**	**Usual Activities**	**Pain/Discomfort**	**Anxiety/Depression**
Additional IR	1.86 ± 0.74	2.03 ± 2.03	2.05 ± 0.73	2.00 ± 0.73	2.06 ± 0.71
Only IR	2.15 ± 0.71	2.06 ± 067	2.04 ± 073	1.99 ± 0.72	2.04 ± 0.72
	*p* < 0.05	ns	ns	ns	ns

**Table 5 medicina-56-00666-t005:** The average scores of the SIS domains baseline, six and twelve months of survivors who underwent only inpatient rehabilitation and among survivors who underwent additional inpatient rehabilitation.

**SIS Domains**	**Period**	**± Survivors**		*** Test and *p***
		**Only Inpatient Rehabilitation** **(*n* = 80)**	**Additional Inpatient Rehabilitation** **(*n* = 80)**	
	** Baseline	26.67 ± 24.19	24.82 ± 21.46	0.611
	6 months	56.25 ± 30.18	61.12 ± 26. 1	0.285
	12 months	60.42 ± 30.76	69.67 ± 27.43	0.046
Memory	Baseline	53.75 ± 23.43	49.72 ± 28.18	0.300
	6 months	74.06 ± 20.34	75.32 ± 24.43	0.708
	12 months	78.23 ± 18.40	80.88 ± 23.16	0.393
Emotions	Baseline	45.88 ± 11.29	49.39 ± 12.51	0.055
	6 months	54.81 ± 10.35	57.07 ± 9.89	0.158
	12 months	55.19 ± 10.70	58.82 ± 9.83	0.027
Communications	Baseline	66.07 ± 25.21	65.49 ± 28.77	0.888
	6 months	89.52 ± 15.48	88.92 ± 19.69	0.817
	12 months	92.38 ± 12.36	91.02 ± 18.74	0.548
Daily activity	Baseline	6.94 ± 9.87	9.34 ± 14.77	0.183
	6 months	27.15 ± 23.69	27.02 ± 27.61	0.973
	12 months	47.57 ± 32.30	51.13 ± 3151	0.475
Mobility	Baseline	36.75 ± 29.36	37.79 ± 28.30	0.817
	6 months	50.08 ± 33.11	61.91 ± 28.08	0.018
	12 months	61.42 ± 33.54	69.96 ± 28.13	0.048
Hand	Baseline	16.00 ± 27.57	13.46 ± 25.04	0.542
	6 months	32.83 ± 37.50	34.49 ± 3.36	0.776
	12 months	38.67 ± 39.91	45.51 ± 41.60	0.277
Participation role	Baseline	5.83 ± 13.17	8.99 ± 16.73	0.158
	6 months	16.48 ± 26.31	28.76 ± 31.22	0.005
	12 months	24.72 ± 32.22	38.44 ± 34.11	0.008

SIS—Stroke Impact Scale. * Mann–Whitney U test; ** one month after discharge.

**Table 6 medicina-56-00666-t006:** Results of multivariant linear regression analyses.

Factor	EQVAS				EQ5D			
	OR	95% CI for OR		*p*	OR	95% CI for OR		*p*
		Lower	Upper			Lower	Upper	
Inpatient rehabilitation-IR	0.054	0.007	0.420	0.005	0.837	0.423	1.657	0.610
Additional IR	3.957	1.913	8.185	<0.001	1.777	1.016	3.107	0.044
Female	0.722	0.368	1.420	0.351	0.924	0.519	1.645	0.787
Age	0.964	0.926	1.626	0.773	1.010	0.979	1.041	0.542

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
