# Peer review of "Health-Related Quality of Life in Stroke Survivors in Relation to the Type of Inpatient Rehabilitation in Serbia: A Prospective Cohort Study"

_medicina, 2020, doi:10.3390/medicina56120666_

Round 1

Reviewer 1 Report

Extensive English language editing required

Author Response

Very Rescpected Rewier,

Thank You very much for Your time. I did the corrections in english.

Reviewer 2 Report

The authors performed a prospective cohort study on stroke survivors with and without additional 6 months inpatient rehabilitation, and demonstrated that there was a significant difference for the two groups of patients in their perceived health-related quality of life. This study is valuable and could provide guidance for the recovering of stroke survivors to improve their quality of life. I suggest this work be published in medicina after minor revision.

Suggested revision list:

  1. line 41-42. Could you comment more or provide more supporting reference regarding why HRQOL measurements are more relevant to patients than measures of impairment or disability? This could make your use of HRQOL measurement for the study more convincing.
  2. line 58. “as significantly decreased HRQOL” is hard to understand. I think part of the sentence is missing.
  3. line 105. “ICU”. You gave the definition of ICU in line 60, no need to repeat it. Similar issue for “SIS” at line 140.
  4. line 236-237. “The results of our study are in agreement.…There are similar results in the literature” Could you be more specific about the results that you mentioned here? If similar results had been reported, then the significance of this study will diminish.
  5. line 240-241. “Twelve months after…SIS domains.” The sentence is hard to understand.
  6. line 288. “Abbreviations” Could you add “EQVAS” and “IR” into this section since the two terms appeared in the content very often.

Author Response

Very respected reviewer,

Thank You very much for Your time.

I correct Enlish.

Reviewer 3 Report

The authors present data regarding health related quality of life in a post stroke cohort which was compared based on standard versus additional inpatient rehab.  The study seems well constructed.  However could the authors clarify in their text the following.

  1. How were patient's assigned to extended inpatient rehab versus not?
  2. The authors demonstrate significant domain changes in mobility and participation role--can the authors speculate as to why these domain's improved significantly i.e. were there any specific activities or programs in additional inpatient rehab that would suggest that these domains would be most affected.  Also can the authors speculate why the other domains did not change.
  3. The authors provide the aggregate scores in terms of health quality of life improvement. could they provide perhaps insight into which domains in the EQ5D changed significantly for standard IR versus additional IR as the authors indicate that only additional IR patients lowered their EQ5D scores from 6 to 12 months post discharge
  4. Is there any other follow up from neurologists or PMR specialists (or other practioners) that can be used to objectively tie EQ5D scores with performance data and the time points used.  This I know would be additional analysis but could provide robust information as to how health quality of life is reflective of observer assessment of clinical improvement.

Author Response

  1. Interdisciplinary team of doctors decided about additional IR accodring to the health condition and other important factors.
  2.  Acivities in the spa were focus predominamtly on the motor recovery.
  3. I added new table with average values of eacf EQd Domain.
  4. I dont know.